# On-the-Fly Adaptation of Source Code Models

**Disha Shrivastava** [1] [2]   **Hugo Larochelle** [2] [1] [3]   **Daniel Tarlow** [2] [4]

## Abstract

The ability to adapt to unseen, local contexts is an important challenge that successful models of source code must overcome. One of the most popular approaches for the adaptation of such models is dynamic evaluation. With dynamic evaluation, when running a model on an unseen file, the model is updated immediately after having observed each token in that file. In this work, we propose instead to approach this problem in two steps: (a) We select targeted information (*support tokens*) from the given context; (b) We use these support tokens to learn adapted parameters which are then used to predict the target hole. We refer to our proposed framework as Targeted Support Set Adaptation (TSSA). We consider an evaluation setting that we call *line-level maintenance*, designed to reflect the downstream task of code auto-completion in an IDE. We demonstrate improved performance in experiments on a large scale Java GitHub corpus, compared to other adaptation baselines including dynamic evaluation. Moreover, our analysis shows that, compared to a non-adaptive baseline, our approach improves performance on identifiers and literals by 44% and 19%, respectively.

## 1. Introduction

Statistical language models for source code (Hindle et al., 2012), like natural language, are usually designed to take as input a window of tokens $w$ and produce a predictive distribution for what the next token $t$ might be. However, factors such as proliferation of vocabulary due to identifiers (such as names of classes, methods and variables) (Karampatsis & Sutton, 2019), occurrence of repetitive patterns in local context (Tu et al., 2014) and faster rate of evolution of software corpora (Hellendoorn & Devanbu, 2017), make modelling source code different from modelling natural lan-

guage. According to Allamanis & Sutton (2013), in the Java GitHub corpus test set, for each project, on an average 56.49 original identifiers (not seen in the training set) are introduced every thousand lines of code. There are also coding styles and conventions that are specific to each file and may not necessarily be seen in the training data. Each organization or project may impose its own unique conventions related to code ordering, library and data structure usage, and naming conventions. Additionally, developers can have personal preferences in coding style (e.g., preferring $j$ as a loop variable to $i$). These motivate us to develop models that adapt their parameters to unseen contexts "on the fly", i.e. they efficiently adapt to test files, even if the file contains identifiers and conventions that were unseen at training time.

A popular approach for model adaptation employed for natural language (Mikolov et al., 2010; Krause et al., 2018) and also advocated for source code (Karampatsis et al., 2020) is dynamic evaluation. With dynamic evaluation, we allow updating the parameters of a trained model on tokens in test files, from the first token to the last. To avoid bias and obtain an unrealistically optimistic measure of performance (i.e. cheating), the prediction of a token in a test file is made before updating the model's parameters.

In this work, to reflect the way a software developer uses auto-completion in an IDE, we consider an evaluation setting that we call *line-level maintenance*. We imagine a cursor placed before a random token in a given file. We blank out the remainder of the line following the cursor to simulate a developer making an in-progress edit to the file. The task is then to predict the token (or *hole target*) that follows the cursor. This setting is different from the language modelling setting, where a test file is generated from scratch one token at a time, from top to bottom. Similarly, dynamic evaluation is ill-suited to this setting, as it processes tokens in that same order. Instead, we propose to select targeted information from both before and after the hole as a basis for adaptation.

In this work, we introduce Targeted Support Set Adaptation (TSSA), which leverages the notion of *support windows* and *support tokens* retrieved "on the fly" at test time. Figure 1 presents the specific task of predicting, on line 20, a hole target $t^h$ from its hole window $w^h$ or preceding tokens. To

---

[1]Mila, Université de Montréal [2]Google Research [3]CIFAR fellow [4]Mila, McGill University. Correspondence to: Disha Shrivastava <dishu.905@gmail.com>.

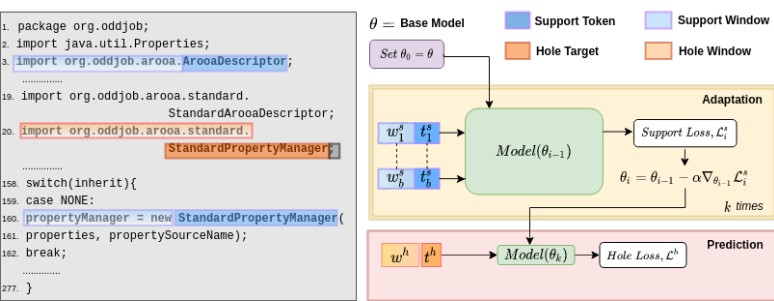

*Figure 1.* **Block diagram illustrating our approach for a sample file**. To predict hole target ***StandardPropertyManager*** using hole window ($w^h$), our model learns parameter $\theta_k$ by performing $k$ steps of gradient update using support tokens ($t^s$) and support windows ($w^s$) in its inner loop.

improve this prediction, in TSSA we leverage support tokens $t^s$ (along with preceding tokens or support window $w^s$), which are tokens from around the file that we believe to be particularly influential in defining the nature of the local context. Intuitively, these could be tokens that are unique to the file and hence provide strong signal for adaptation. The inner loop predicts support tokens $t^s$ from support windows $w^s$ and takes multiple gradient steps to update the parameters of the source code model and reduce the loss of its predictions. The updated parameters are then used to predict the hole target $t^h$ from the hole window $w^h$. Our contributions can be listed as follows:

- We introduce TSSA, which formulates the problem of adaptation to local, unseen context in source code by retrieving targeted information (support tokens) from both before and after the hole in a file. (Section 3.2.2).

- We consider a new setting that we call line-level maintenance for evaluating models for source code in a way that is directly inspired by the way developers operate in an IDE (Section 3.1).

- Via experiments on a large-scale Java GitHub corpus, we demonstrate that TSSA significantly outperforms baselines including dynamic evaluation, even with half the number of adaptation steps. Further, via ablations we show that we improve performance on identifiers and literals by about 44% and 19% respectively (Section 4.3).

## 2. Related Work

There have been numerous efforts in developing models for source code, such as $n$-gram based (Hindle et al., 2012; Nguyen et al., 2013), CRF-based (Raychev et al., 2015; Bichsel et al., 2016), probabilistic graphical model based (Maddison & Tarlow, 2014; Raychev et al., 2016; Bielik et al., 2016); and Neural-networks based (White et al., 2015; Allamanis et al., 2018; Dam et al., 2016).

Some of these focus specifically on code-completion applications (Raychev et al., 2014; Alon et al., 2019; Svyatkovskoy et al., 2020; Li et al., 2018; Wang et al., 2020; Svyatkovskiy et al., 2020). To tackle the specific challenge of local context adaptation Tu et al. (2014) combined an $n$-gram with the concept of a cache. Later, Hellendoorn & Devanbu (2017) extended this idea to develop nested $n$-gram models combined with a cache. The components in the cache could then come not only from the current file, but also other files in the directory or project, leading to significant improvements in performance. This idea could be adapted to our setting, by collecting support tokens beyond just the current file. Follow up work from Karampatsis et al. (2020) have established the current state-of-the-art. They use deep recurrent models based on subword units. They apply dynamic evaluation by performing updates using information from all the files in a project and carrying over the updated value of parameters from one test file in the project to another during evaluation. However, on average this results in a long chain of adaptation steps before a prediction is made, which may present challenges when deploying in a real IDE (e.g., how to do quality control when the parameters used in the deployed system won't be known at release time?). In this work, we instead focus on and perform controlled experiments in a single file setting with a much smaller number of allowed update steps, which is more generally applicable.

## 3. Methodology

### 3.1. Line-level Maintenance

The line-level maintenance task is both more realistic (developers typically edit files rather than generating them from left-to-right) and creates the need for stronger forms of adaptation. More concretely, we refer to a file $f$ as a sequence of tokens $t_1, t_2, ....t_N$. As per Karampatsis & Sutton (2019), we represent each token $t_n = (s_1, s_2..., s_{l_n})$ as a list of $l_n$ subtokens. Our task is to predict the first token (called *hole target*) in the blanked out range, which occurs at a partic-

ular position in the file. For an example, refer to Figure 1 where the hole target is highlighted in dark orange and the blanked out range is highlighted in black. Note that we are not allowed to use any token from the blanked-out range.

## 3.2. Adaptation

### 3.2.1. BASE MODEL

We begin by defining a *base model*, which is a Seq2Seq (Sutskever et al., 2014) model trained to predict the sequence of subtokens in the hole target $t^h$ from the sequence of subtokens in the hole window $w^h$ using parameters $\theta$. The probability of hole target given its window can be written as

$$p(t^h|w^h;\theta) = \prod_{s_i \in t^h} p(s_i|s_{i-1}, ..., s_1, w^h; \theta). \quad (1)$$

During training of the base model, each token in the file is used as a hole target.

### 3.2.2. TARGETED SUPPORT SET ADAPTATION (TSSA)

To adapt the base model to the local file context, we consider regions from the file that potentially provide useful cues for predicting a given hole target. We call this set of tokens and preceding windows the *support set*, inspired by the usage of the term in few-shot learning (Vinyals et al., 2016). Each element of the support set, $S = \{(w^s, t^s)\}$ is a pair of support window $w^s$ and support token $t^s$. The support windows and support tokens can come from anywhere in the file except for the blanked out remainder of the line following the hole target.

To adapt the model given a support set, we perform $k$ steps of gradient descent over each of the $k$ mini-batches of support windows and tokens. In each step, we predict the support token from the corresponding support window using the base model with parameters from the previous step. The *support loss* at step $i$ and the updated parameters at step $i$ can be written as

$$\mathcal{L}_i^s = \frac{1}{b} \sum_{j=1}^{b} \log p(t_{ij}^s|w_{ij}^s; \theta_{i-1}) \quad (2)$$

$$\theta_i = \theta_{i-1} - \alpha \nabla_{\theta_{i-1}} \mathcal{L}_i^s \quad \text{[Inner Update]}, \quad (3)$$

where $i \in \{1, \dots, k\}$, $\theta_0 = \theta$, $b$ = mini-batch size and $\alpha$ = hyperparameter corresponding to the inner adaptation learning rate. We then use the updated parameters $\theta_k$ to predict the hole target from its hole window, resulting in the *hole loss* $\mathcal{L}^h$

$$\mathcal{L}^h = \log p(t^h|w^h; \theta_k). \quad (4)$$

### 3.2.3. SUPPORT SET SELECTION STRATEGIES

A key novelty in this work is the idea of actively choosing a support set that leads to effective adaptation. This is in contrast to, e.g., few-shot learning, where the support set is defined by the task and cannot be changed. We can think of it being similar to self-supervised learning in the sense that the tasks are created from the given context.

In source code, identifiers are the most difficult to predict (Allamanis & Sutton, 2013) and also the most frequent of all token-types (Broy et al., 2005), making it the most common use-case for auto-complete systems. Thus, our definitions of support tokens are aimed at providing additional context that should help in predicting identifiers. We are motivated by the fact that identifiers are frequently re-used within a file even if they are uncommon across files (or even if they only appear in one file). Further, even when there is not an exact match, it is common for there to be repeated substructure in identifiers. Our work offers advantage compared to just using a powerful base model, like a transformer which has fixed context window size around the target hole and hence is ineffective to make use of these patters which are far away from the cursor in the current file, especially if the file is long.

With this in mind, we explored four definitions of support tokens (which contribute towards determining the support sets): (a) Vocab: Tokens that are rare in the corpus; (b) Proj: Tokens that are relatively common in the current project but are rare in the rest of the corpus; (c) Unique: Single occurrence of a token in the support set; and (d) Random: Tokens are randomly selected. More details about each of these can be found in Appendix A.

## 4. Experiments and Results

### 4.1. Experimental Details

For our experiments, we work with the Java GitHub Corpus provided by Allamanis & Sutton (2013). All our models are Seq2Seq networks where both encoder and decoder networks are recurrent networks with a single layer of 512 GRU (Cho et al., 2014) hidden units, preceded by a trainable embedding layer of equal size. To train the base model, we create minibatches of successive target holes as in standard training of language models, and we train to minimize average token loss. We use mini-batches of support tokens and the Adam (Kingma & Ba, 2015) optimizer in the adaptation inner loop. An important note is that during evaluation, at the beginning of each inner loop execution, we not only set $\theta_0$ to $\theta$, but also set the state of the Adam optimizer to its value from the end of training. The latter step ensures that the statistics for Adam are not carried from one file to another. Details about the dataset and preprocessing; and best hyperparameter values for all settings can be obtained

| Model | Cross Entropy | MRR@10 (All)(%) | MRR@10 (Identifiers)(%) | Recall@10 (All)(%) | Recall@10 (Identifiers) (%) |
|---|---|---|---|---|---|
| Base Model | $5.222 \pm 0.10$ | $65.20 \pm 0.42$ | $24.90 \pm 0.64$ | $75.74 \pm 0.42$ | $36.20 \pm 0.78$ |
| Dynamic Evaluation | $3.540 \pm 0.08$ | $\mathbf{68.95 \pm 0.41}$ | $34.44 \pm 0.70$ | $80.39 \pm 0.39$ | $48.86 \pm 0.82$ |
| TSSA-1 | $3.461 \pm 0.07$ | $66.94 \pm 0.40$ | $35.76 \pm 0.70$ | $81.00 \pm 0.38$ | $52.04 \pm 0.82$ |
| TSSA-8 | $3.383 \pm 0.06$ | $67.52 \pm 0.40$ | $35.14 \pm 0.70$ | $80.65 \pm 0.38$ | $50.27 \pm 0.82$ |
| TSSA-16 | $\mathbf{3.240 \pm 0.06}$ | $68.63 \pm 0.40$ | $\mathbf{36.74 \pm 0.70}$ | $\mathbf{81.51 \pm 0.38}$ | $\mathbf{52.34 \pm 0.82}$ |

*Table 1.* **Performance on hole target prediction on test data in terms of token cross-entropy, MRR@10 and Recall@10**. We also report 95% confidence intervals for each entry. We highlight the best performing models (in terms of mean) for each column.

from Appendix B and Appendix C, respectively.

### 4.2. Evaluation Setup

There is a trade-off between accuracy and number of inner loop updates of adaptation. More inner loop updates generally improve cross-entropy but come at the cost of computation time and ultimately latency in a downstream autocomplete application. To control for this, we fix the size of batches and number of updates per hole target prediction across all adaptive methods. We measure the performance of our models in terms of token cross-entropy, MRR@10 and Recall@10 (see Appendix E for details on these metrics). We experimented with the following methods:

**1. Base model**: This is the pretrained base model used as is, without any contextual adaptation. This comparison allows us to confirm the benefit of adaptation in general.
**2. TSSA-$k$**: This corresponds to doing $k$ steps of inner loop adaptation using support tokens. We also report results for TSSA-1 (single inner-loop update), to highlight the value of multiple updates.
**3. Dynamic Evaluation**: We also implement dynamic evaluation in our framework which is a bit different from in Karampatsis et al. (2020). Here, 1) the support sets are made of all window/tokens pairs $(w^s, t^s)$ appearing *before* the hole target (and none after), and 2) we constrain the inner-loop optimization to order its updates by starting at the beginning of the file, until the token right before the hole target. Thus, the first inner-loop mini-batch of size $b$ contains tokens at the beginning of the file, while the tokens immediately before the hole target only appear in the last mini-batch. Moreover, if the hole target is the $m$th token in the file, then there will be $ceil(m/b)$ updates in total. The variants of TSSA assume a fixed number $k$ of inner-loop updates, unlike dynamic evaluation. To allow for an overall fair comparison, we set $k$ to the average number of updates performed by dynamic evaluation, which was found to be approximately 16 for our test data.

### 4.3. Results

In Table 1, we report the average cross-entropy, MRR@10 and Recall@10 for test hole targets (all token types and identifiers). In these results, we sample five holes per file to measure test performance. For each method, we select the

best values of hyperparameters using the performance on the validation data. As can be seen from the table, TSSA-16 gives the best performance in terms of cross-entropy, MRR(Identifiers) and Recall; and is comparable to dynamic evaluation in terms of MRR (all). It is interesting to note that even TSSA-1 and TSSA-8 outperform dynamic evaluation in terms of cross-entropy, MRR(Identifiers) and Recall; even though they perform significantly less adaptation steps (single and half the number of adaptation steps, respectively as compared to dynamic evaluation (16)). This huge saving in terms of computational cost, is especially attractive while deploying models in an IDE where low latency is required.

We also analysed how our framework performs with hole targets of different token-types (See Appendix F and Appendix D). We found that identifiers and literals (string literals, char literals, etc.) are the most difficult to predict amongst all token types. Table 2 shows the comparison of average test cross-entropy values for the non-adaptive base model as compared to our best model (TSSA-16). As can be seen from the table, we obtain significant reduction in cross-entropy values of about 44% and 19%, respectively in case of identifiers and literals. This in turn leads to better performance overall. See Appendix for results of ablation studies, sample cases and TSSA vs bigger model comparisons.

| Token Type | Base model | TSSA-16 | % Improvement |
|---|---|---|---|
| Identifiers | 13.16 | 7.35 | 44.15 |
| Literals | 7.18 | 5.82 | 18.94 |

*Table 2.* **Comparison of cross-entropy on prediction of identifiers and literals for TSSA-16 vs. a non-adaptation model.**

## 5. Conclusions

In this work, we propose TSSA: an approach which selects targeted information from the local context and then uses this to learn adapted parameters, which can then be used for predicting a hole target in the current file. Our experiments on a large-scale Java GitHub corpus reveal the following: (a) Our formulation significantly outperforms all baselines including a comparable form of dynamic evaluation, even with significantly less adaptation steps in many cases; (b) Most of our performance benefits comes from reducing the cross-entropy on identifiers and literals. For future, we want to learn the criteria for building support sets.

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

## A. Support Set Definitions

In all cases, we ensure that the selection of support sets does not depend on the hole target or the blanked out region following the hole target.

1. **Vocab:** We try to capture tokens that are rare in the corpus as part of support tokens. We take all the tokens from the file and sort them based on their frequency in the vocabulary in reverse order and then take the top-$N$ entries.

2. **Proj:** Here, as part of support tokens, our target is to capture tokens that are relatively common in the current project but are rare in the rest of the corpus. We divide each token's frequency in the project with the frequency in the vocabulary, sort them and then take the top-$N$ entries.

3. **Unique:** To study if multiple occurrences of the same token in the support set helps, we form a set of tokens in the file. We then take a subset of $N$ tokens as part of our support set. Here, each support token in the support set is unique.

4. **Random:** We take $N$ random tokens from the file as support tokens.

## B. Dataset and Preprocessing

We work with the Java GitHub Corpus provided by Allamanis & Sutton (2013). It consists of open-source Java repositories for more than 14000 projects. Java is a convenient choice as it is one of the most popular languages for software development and has been widely used in previous works (Karampatsis & Sutton, 2019; Tu et al., 2014). Following Hellendoorn & Devanbu (2017), we focus on a 1% subset of the corpus. The name of the projects in training, validation and test splits of the dataset were taken from Hellendoorn & Devanbu (2017)[1]. Statistics of the data are provided in Table 3. Note that while we show results on Java, our method is otherwise applicable to corpora of any programming language.

| Feature | Train | Val | Test |
|---|---|---|---|
| # Projects | 107 | 36 | 38 |
| # Files | 12934 | 7185 | 8268 |
| # Lines | 2.37M | 0.50M | 0.75M |
| # Tokens | 15.66M | 3.81M | 5.31M |
| # Identifiers | 4.68M | 1.17M | 1.79M |

*Table 3.* **Corpus Statistics for 1% split of the dataset**. M indicates numbers in millions

We made use of the lexer provided by Hellendoorn & Devanbu (2017)[1] to tokenize the files, preserving line-breaks. Note that the lexer also removes comments in the file. We need to use a Java-specific tokenizer because characters such as dot or semi-colon take a special meaning in Java and are not tokenized as individual tokens by NLP parsers. To get the Java token-types, we made use of Python's Java-parser.[2] Subword tokenization was performed using the subword text encoder provided by Tensor2Tensor (Vaswani et al., 2018). As in Karampatsis & Sutton (2019), we use a separate vocabulary data split, consisting of a set of 1000 randomly drawn projects (apart from the projects in 1% split), to build the subword text encoder. In addition, we append an extra end-of-token symbol (EOT) at the end of each Java token. The final size of the subword vocabulary is 5710.

## C. Details of Hyperparameter Values

In all settings of our Seq2Seq Models, the initial decoder state is set to be the last state of the encoder. The first input to the decoder is the last step output of the encoder. A dense layer with softmax output is used at the decoder. Also, note that both the parameters of the model and the state of Adam is reset after each hole target during evaluation. We use a dropout = 0.5 and gradient clipping = 0.25. We embedding layer dimension is equal to the hidden layer dimension = 512. We take both the support and hole window size to be 200. In Table 5 we define the best hyperparameter values for all our settings. Notation for reading Table 5 is provided in Table 4. For our experiments, we use NVIDIA P100 and K80 GPUs with 16GB memory each. To reduce model computation while decoding, we remove hole targets of length greater than or equal to 20 subwords. These constitute only 0.2% of the total number of tokens in training data and 0.1% in validation and test data, making it less significant.

---

[1]https://github.com/SLP-team/SLP-Core

[2]https://pypi.org/project/javac-parser/

| Symbol | Meaning |
|--------|---------|
| lr | learning rate of Adam optimizer |
| hbs | hole batch-size |
| dbs | batch-size of tokens in dynamic evaluation |
| sbs | support tokens batch-size |
| #up | number of inner loop updates |
| snum | number of support tokens |
| sdef | definition of support tokens |
| ilr | learning rate of inner update Adam optimizer |
| T: | while training/ meta-training |
| E: | while evaluation |

*Table 4.* **Notation for terms occurring in Table 5**

| Model | Hyperparameters |
|-------|-----------------|
| T: Base Model | lr = 1e-4, hbs = 512 |
| E: Base Model | hbs = 1 |
| E: Dynamic Evaluation | lr = 1e-3, , hbs = 1, dbs = 20 |
| E: TSSA-1 | lr = 5e-3, hbs = 1, sdef = proj, snum = 1024 |
| E: TSSA-8 | lr = 1e-3, hbs = 1, sbs = 20, sdef = vocab, #up = $k$ = 8, snum = 256 |
| E: TSSA-16 | lr = 5e-4, hbs = 1, sbs = 20, sdef = vocab, #up = $k$ = 16, snum = 256 |

*Table 5.* **Best hyperparameter values for all our settings**

## D. Categorization of Token Types

| Token Category | Java Token-Type |
|----------------|-----------------|
| Identifiers | identifier |
| Keywords | import, break, throws, extends, for, public, return, protected, boolean, package, new, class, void, static, int, this, volatile, synchronized, if, private, final, implements, super, catch, try, throw, else, instanceof, long, abstract, enum, case, byte, char, break, interface, finally |
| Operators | dot, gt, lt, eq, plus, eqeq, colon, bangeq, ques, ampamp, sub, bang, plusplus, barbar, star, amp, gteq, subsub, bar, ellipsis |
| Literals | stringliteral, intliteral, charliteral, longliteral, null, false, true |
| Special Symbols | semi, rparen, lparen, lbrace, rbrace, comma, monkeys_at, rbracket, lbracket |

*Table 6.* **Description of Java token-types given by Python's Java-parser into broad token categories for ease of visualization**

## E. Evaluation Metrics

- **Cross-Entropy.** It is the average negative log probability of tokens, as assigned by the model. It rewards accurate predictions with high confidence and also corresponds to the average number of nats required in predicting a token. The cross-entropy of a sequence $T$ with probability $p(T)$ under a model, is:

$$H_p(T) = -\frac{1}{m} \log p(T) \tag{5}$$

  We evaluate the average under a distribution over hole target tokens where we first sample a file uniformly from the set of all files and then sample a hole target token uniformly from the set of all tokens in the file. This reflects the assumption that a developer opens a random file and then makes an edit at a random position in the file.

- **MRR/ Recall:** Since our approach can be used for code-completion (predicting the hole target), we need some metrics

to measure the accuracy at this task. Mean Reciprocal Rank (MRR@$n$) is the average of the inverse of the position of the correct answer in a ranked list of size $n$. Recall@$n$ is 0 or 1 based on the absence or presence of the correct answer in the ranked list of size $n$.

## F. Performance across Token-Types

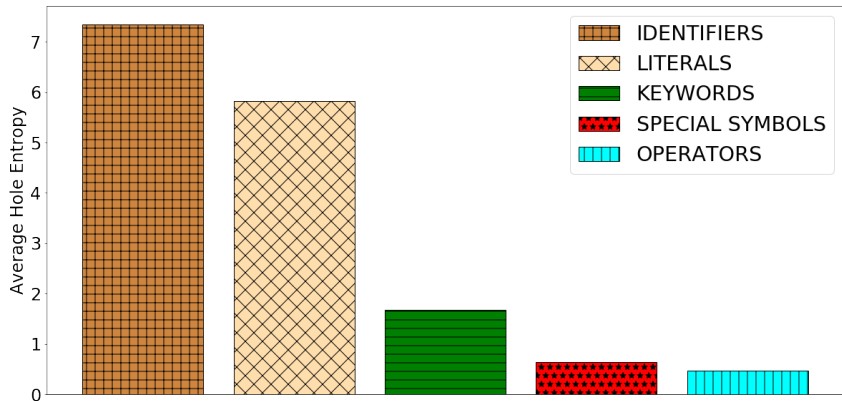

*Figure 2.* **Average hole target cross-entropy for each token-type for our TSSA-16 model**

## G. Ablation Studies

In this section, we try to draw insights into the workings of our framework by analyzing the role of each component. We took our best performing TSSA-16 for all the experiments that follow. In Figure 3, we plot the variation of hole target cross-entropy values with the number of updates and number of support tokens ($N$ from Section **??**), for validation data. As can be seen from the plot, the cross-entropy decreases with more updates. We also see that for a fixed number of updates, the cross-entropy decreases with the number of support tokens only until it reaches a certain point after which it increases. This likely arises from the way we form mini-batches of support tokens where we first shuffle the support tokens and then cycle through them until exhausting the number of updates. This suggests that going past the point where each support token has been visited once creates redundancy that is detrimental.

We also experimented with the definition of support tokens where in one case we fixed the number of updates (16), while in the second we fixed the number of support tokens (256). Figure 4 displays the results for validation data. We see that the *Vocab* definition of support tokens performs best closely followed by *Proj*. On the other hand, *Unique* and *Random* perform worse in both cases. This highlights the fact that how we define support tokens indeed plays a role in performance improvement.

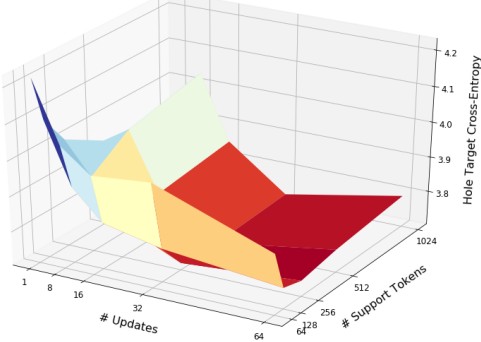

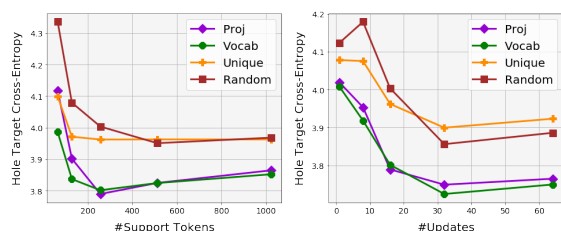

*Figure 3.* **Variation of hole target cross-entropy values with number of updates and number of support tokens for val data**

*Figure 4.* **Variation of token cross-entropy for val data with different definition of support tokens**. (***Left***) With fixed number of updates; (***Right***) With fixed number of support tokens.

## H. TSSA vs. Bigger Model

One question is if benefits gained from TSSA are similar to or orthogonal to benefits that would arise from using larger and more sophisticated models. To study this question, we start from a "small base model" (256 hidden units) and build two models that improve, but in different directions. The first "big base model" increases the model size to 512 hidden units. The second "small TSSA" model leaves the hidden sized fixed but employs TSSA-16. We then compare how individual examples benefit from each kind of modelling improvement. Specifically, let the hole target cross-entropy for the small base model be $b_{low}$, for the big base model be $b_{high}$, and for the small TSSA model be $m_{low}$. In the right part of Figure 5 we plot the improvement obtained due to higher capacity model $b_{low} - b_{high}$ on the x-axis and improvement due to the low-capacity meta-learnt model $b_{low} - m_{low}$ on the y-axis. Each point represents a different test hole target. The line marks cases where improvement from both models is equal. First, we see that the majority (57.7%) of the points are above the line, indicating that applying TSSA improves on more cases than increasing the model size. Second, and perhaps more interestingly, there are many points where the improvement due to increasing model size is near zero, indicating that we have achieved saturation in benefit due to increasing model size in these cases. However, using TSSA here, even with the small model, often leads to a large improvement in performance. This shows that TSSA can help in adapting even when we reach saturation in terms of model capacity.

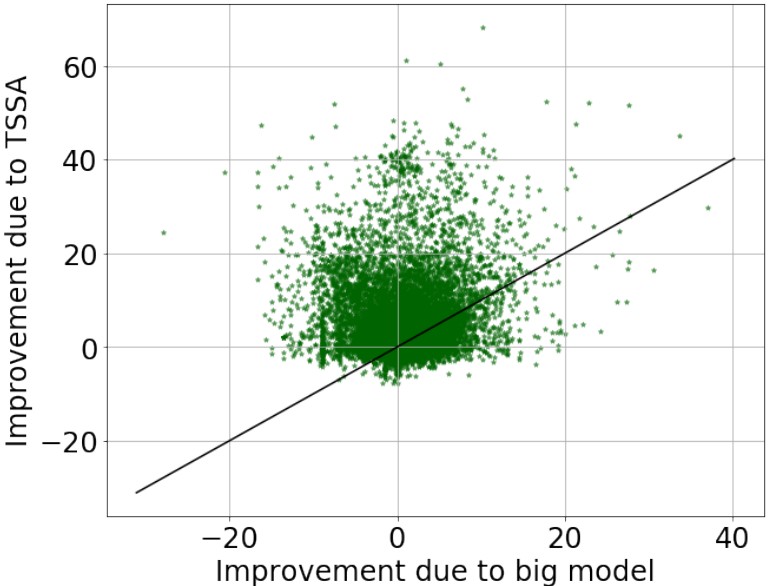

*Figure 5.* Improvement due to TSSA on small capacity model ($b_{low} - m_{low}$) vs. Improvement due to big model ($b_{low} - b_{high}$)

## I. Sample Cases

In Figure 6 we showcase two such sample cases. For the left one, we have a string literal as hole target (***"column("***). We can see that fragments of it can be found in support tokens (highlighted in blue). The right one has an identifier (***WGLOG***) as hole target. Somewhere far later in the file, we find a support token that exactly matches the hole target, contributing to a large gain in performance of TSSA as compared to no adaptation. In neither of these cases does a larger or more sophisticated base model help in harnessing this extra information.

```
1. package de.fuberlin.wiwiss.d2rq.mapgen;
2. public class FilterMatchColumn extends Filter{
3. private final IdentifierMatcher schema;
   ............
34. public String toString(){
35. StringBuffer result = new
                          StringBuffer("column(");
36. if(schema != Filter.NULL_MATCHER){
   ............
42. result.append(column);
43. result.append(")");
44. return result.toString();
45. }
46. }
```

```
1. package com.asakusafw.windgate.retryable;
2. import java.io.IOException;
3. import java.text.MessageFormat;
   ............
31. public class RetryableProcessProfile {
32. static final WindGateLogger WGLOG = new
                   RetryableProcessLogger( ..........
33. private static final char SEPARATOR = '.';
   ............
91. } catch (Exception e) {
92. WGLOG.error(e, "E00001",
93. profile.getName(),
   ............
```

*Figure 6.* **Sample cases illustrating the benefits of TSSA on low capacity model:** *(Left)* Hole target is string literal with partial match in support tokens; *(Right)* Hole target is identifier with exact match in support tokens.