# OpenReview forum: "On-the-Fly Adaptation of Source Code Models"
_NeurIPS.cc/2020/Workshop/CAP — NeurIPS 2020 CAP Workshop_

### Official Review · AnonReviewer1 · 2020-10-30
**Interesting problem and good empirical results**

**Rating:** 6
**Confidence:** 4

**Review:**

This paper studies the problem of dynamically adapting a code completion model to the current document. In particular, unlike NLP tasks, in code completion there are many document-specific properties such as variable names or code styles that cannot be learned from training data. Thus, the goal is to dynamically tune the model to the current context. While existing approaches exist, they are not suited to the non-sequential nature of the code completion problem.

The paper proposes a simple but effective technique: train on a small set of “support tokens” chosen from the current document based on several heuristics. They compare to baselines, including existing technique “dynamic evaluation” from the NLP domain that dynamically adapts the model by fine-tuning it on the available portion of the document.

Overall, I think this paper studies an interesting problem. While their approach is a bit simplistic, it appears to work well and they obtain good empirical results.

The authors claim their approach performs better than the dynamic evaluation baseline because they are specific to the sequential nature of the model. One question I have is whether tree-LSTM variants of existing techniques would apply better than the sequence-LSTM. I suspect the authors approach would still work better due to the targeted heuristics, but it would be an interesting comparison nonetheless.

---

### Decision · Program_Chairs · 2020-11-03

**Decision:**

Accept

**Comment:**

The PCs have decided to accept this paper.